# A Ground-Based Electrostatically Suspended Accelerometer

**DOI:** 10.3390/s24124029

**Published:** 2024-06-20

**Authors:** Hanxiao Liu, Xiaoxia He, Chenhui Wu, Rong Zhang

**Affiliations:** 1Department of Precision Instrument, Tsinghua University, Beijing 100084, China; liu-hx23@mails.tsinghua.edu.cn (H.L.); rongzh@tsinghua.edu.cn (R.Z.); 2State Key Laboratory of Precision Space-Time Information Sensing Technology, Beijing 100084, China

**Keywords:** ground-used accelerometer, electrostatic suspension, proof mass structure, gravity detection

## Abstract

In this study, we have developed an electrostatically suspended accelerometer (ESA) specifically designed for ground use. To ensure sufficient overload capacity and minimize noise resulting from high suspension voltage, we introduced a proof mass design featuring a hollow, thin-walled cylinder with a thin flange fixed at the center, offering the highest surface-area-to-mass ratio compared to various typical proof mass structures. Preload voltage is directly applied to the proof mass via a golden wire, effectively reducing the maximum supply voltage for suspension. The arrangement of suspension electrodes, offering five degrees of freedom and minimizing cross-talk, was designed to prioritize simplicity and maximize the utilization of electrode area for suspension purposes. The displacement detection and electrostatic suspension force were accurately modeled based on the structure. A controller incorporating an inverse winding mechanism was developed and simulated using Simulink. The simulation results unequivocally demonstrate the successful completion of the stable initial levitation process and suspension under ±1g overload.

## 1. Introduction

The Electrostatically Suspended Accelerometer (ESA) is an inertial instrument that relies on electrostatic suspension, primarily consisting of electrodes and a proof mass. Within a high-vacuum electrode housing, the proof mass is suspended solely by electrostatic forces, ensuring no mechanical contact with the surroundings. By applying control voltage to the electrodes, precise manipulation of the proof mass’s position and orientation is achieved, effectively stabilizing it at its equilibrium position. In principle, the electrostatic force can generate subtle but precise accelerations, while capacitive displacement circuits offer exceptionally high positional resolution while minimizing reactive effects. Consequently, electrostatic accelerometers excel at achieving extremely high resolution and high sensitivity.

Due to its remarkable precision potential, especially in low overload environments [1], the electrostatically suspended accelerometer has found extensive use in microgravity space scenarios [2], including equivalence principle experiments [3], gravitational wave detection [4,5], and the test of geodetic and frame-dragging effects [6,7].

In Earth’s gravitational field, electrostatically suspended inertial instruments such as gyroscopes and ESA also exhibit high precision potential. Thus, they have been developed for gravity gradient measurements and gravity-assisted navigation [8,9,10,11]. In the aerospace domain, gravitational field information facilitates the precise navigation of spacecraft [12]. In disaster prevention, the time-variable gravity observed aids in predicting natural disasters such as earthquakes and tsunamis [13]. Currently, gravity measurement methods fall into three main categories: terrestrial static gravity measurement, space-based gravity measurement, and sea-based/airborne gravity measurement. Among these, the terrestrial static gravity measurement offers the highest precision but is inefficient and heavily constrained by terrain. The satellite gravity measurement provides a broader perspective but is limited in its ability to capture high-frequency information in marine regions. In contrast, airborne and sea-based gravimeters offer advantages in terms of cost-effectiveness, efficiency, and freedom from terrain restrictions [14]. Recent advancements in gravity-assisted navigation, including high-precision passive gravity navigation and gravity matching-based seabed terrain estimation techniques [15], signify a progressive trajectory in research efforts.

For sea-based and airborne gravity measurement applications, as well as the emerging field of gravity-assisted navigation, there is a growing need for high-precision ground-based inertial accelerometer components. Ground-based electrostatic accelerometers face more extensive range requirements, a wider operating bandwidth, and the necessity for higher support voltages to achieve levitation and suspension in all directions compared to their space application counterparts. However, the high voltage introduces significant circuit noise, thereby constraining the accuracy of displacement detection systems. Although some efforts have attempted to address this issue using methods such as translation-tilt compensation pendulum bench [16] and low noise circuit [17], the most straightforward and effective approach remains to find a proof mass structure with a surface area to mass ratio as large as possible, boasting excellent mechanical properties, and easily achievable high-precision machining dimensions.

In this paper, we present a structure of proof mass with a significant area-to-mass ratio, along with the corresponding design of an electrostatic force actuation control scheme. By leveraging a proof mass structure with a large area-to-mass ratio, we can reduce the voltage needed for electrostatic suspension, consequently mitigating electric circuit noise and enhancing measurement resolution. Furthermore, we propose a control scheme to manage both the levitation process and suspension stage effectively, and have devised a controller incorporating an inverse “winding” branch to tackle the “winding” phenomenon experienced during the initial levitation process.

The design is explained in the following parts: In Section 2, the structural design and operation principles are described in detail. In Section 3, displacement detection is modeled. Section 4 discusses the electrostatic suspension model and the calculation of electrostatic force. The control schemes and simulation results for the initial levitation and suspension stages are presented in Section 5. Finally, Section 6 concludes this paper.

## 2. Structural Design and Operation Principles

### 2.1. System Overview

The proposed electrostatically suspended accelerometer is a type of closed-loop force balance accelerometer, comprising primarily five components: the proof mass, the electrode distribution structure, the displacement detection circuit, the digital controller, and the voltage amplifier. The operation principle of the accelerometer in one degree of freedom (DOF) is illustrated in Figure 1.

In the absence of external acceleration input, the proof mass is suspended at the system’s equilibrium position by the electrostatic force generated by electrodes. When the external acceleration changes, the proof mass generates displacement x relative to the electrode plates due to inertial force. This displacement triggers a capacitance difference, causing a change in the output $U_{d}$ of the displacement detection circuit. Through a proper controller algorithm, a control signal $U_{c}$ is fed into the high-voltage amplifier, driving an actuation voltage $U_{b}$ apply electrostatic force to the proof mass. By adjusting the electrostatic force magnitude, the proof mass is pulled back to its equilibrium position. At this point, the electrostatic force generated by the actuation voltage $U_{b}$ is equal to the inertial force, and its magnitude and polarity can accurately denote the external acceleration [18].

### 2.2. Designed Structure

Currently, spherical, rectangular, and hollow regular hexahedral structures have been reported for the shape of the proof mass. However, due to their low surface area-to-mass ratio, higher support voltages are required, leading to significant circuit noise.

To address this issue, we propose a proof mass in the form of a hollow, thin-walled cylinder with a thin flange fixed at the inner center, as depicted in Figure 2f. Several possible structures are also shown in Figure 2a–e for comparison purposes. To facilitate a more intuitive comparison of different proof mass structures, the surface area-to-mass ratio values are calculated for these structures and presented in Table 1.

In Table 1, the hollow cylinder with an inner flange structure stands out with the highest surface area-to-mass ratio of 40.3, attributed to its embedded flange. While this proposed structure’s ratio slightly exceeds those of other hollow cylinder designs at 40.1 and 39.9, the inner flange design offers benefits such as simplified fabrication and minimized deformation. This hollow structure design allows for the application of electrostatic force on both the inner and outer cylindrical surfaces, maximizing surface area utilization and aiding in reducing circuit noise. Consequently, the surface area-to-mass ratio is significantly enhanced compared to alternative structures such as hexahedrons and spheres. Our proposed structure, detailed in Figure 3, is capable of achieving a five-degree-of-freedom (DOF) suspension with lower support voltage requirements, owing to its axis-symmetry and high surface area to mass ratio. Currently, the designed prototype is planned to be used in a stable laboratory, so a pinhole is incorporated into the flange to constrain rotational DOF for simplicity. What should be noted is that when it is further used in a rotational vehicle, the “pin collide with the hole” phenomenon will degrade the accelerometer’s precision, and additional electrodes to restrict the rotational DOF should be adopted. The parameters of the proposed structure are presented in Table 2.

To maximize surface area utilization and achieve DOF suspension, the electrodes are distributed on both the inner and outer surfaces of the proof mass, as depicted in Figure 4.

The definitions for the disk electrodes $C_{1}$ and $C_{1}^{\prime }$ and inner and outer cylinder electrodes $C_{\mathrm{in}}$ and $C_{\mathrm{out}}$ are illustrated in Figure 5.

To minimize the maximum suspension voltage, all electrodes are simultaneously utilized for detecting displacement and applying electrostatic support voltage at different frequencies. Additionally, the electrodes for different DOFs are independent of each other, simplifying the control strategy and reducing cross-talk while also reducing the required maximum voltage on each electrode. The assignment of detection and suspension voltage application electrodes for each DOF is outlined in Table 3.

## 3. Displacement Detection of Proof Mass

The displacement detection circuit significantly impacts the accuracy of ESA [19], and some previous studies have thoroughly investigated this matter [20,21]. In this section, we present a displacement detection model that characterizes the relationship between displacement changes/angle changes and the corresponding variation in capacitance due to gap changes. According to Table 2, the displacement detection scheme for three translations and two rotational DOFs is presented in Figure 6. Due to the symmetry of the X and Y DOF, as well as the θ and ϕ DOF, only the displacement detection schemes for the X and θ degrees of freedom are discussed in the following text.

By using the parallel-plate and cylindrical capacitance formulas, the corresponding differential capacitances are deduced as follows:(1)$$\Delta C_{z}=C_{1}-C_{1}^{\prime }=\frac{\varepsilon_{0}\varepsilon_{r}A_{z}}{d_{0z}-z}-\frac{\varepsilon_{0}\varepsilon_{r}A_{z}}{d_{0z}+z}$$
(2)$$\Delta C_{x}=C_{\mathrm{out}1}-C_{\mathrm{out}3}=\varepsilon_{0}\varepsilon_{r}r_{\mathrm{out}}l_{\mathrm{out}}\int_{-\frac{\varphi_{p}}{2}}^{\frac{\varphi_{p}}{2}}(\frac{1}{d_{0}-x\cos\varphi }-\frac{1}{d_{0}+x\cos\varphi })d\varphi$$
(3)$$\Delta C_{\theta }=(C_{\mathrm{in}2}-C_{\mathrm{in}4})+(C_{\mathrm{in}4}^{\prime }-C_{\mathrm{in}2}^{\prime })=2\int_{-\frac{\varphi_{p}}{2}}^{\frac{\varphi_{p}}{2}}\int_{0}^{l_{\mathrm{in}}}(\frac{\varepsilon_{0}\varepsilon_{r}r_{\mathrm{in}}}{d_{0}-ltan\theta \cos\varphi }-\frac{\varepsilon_{0}\varepsilon_{r}r_{\mathrm{in}}}{d_{0}+ltan\theta \cos\varphi })dld\varphi$$
where $d_{0z}$ and $d_{0}$ are the nominal gaps between proof mass and electrodes, r represents the inner and outer diameters of the cylinder, l represents the length of the electrode, and x,y,z,θ,ϕ represent the displacement, which is a small quantity when the accelerometer operates near its equilibrium position. Ignoring second and higher-order quantities, we have:(4)$$\Delta C_{z}=2\varepsilon_{0}\varepsilon_{r}A_{z}\frac{z}{d_{0z}^{2}}=K_{cz}z$$
(5)$$\Delta C_{x}=\frac{4\varepsilon_{0}\varepsilon_{r}r_{\mathrm{out}}l_{\mathrm{out}}}{d_{0}^{2}}\sin(\frac{\varphi_{p}}{2})x=K_{cx}x$$
(6)$$\Delta C_{\theta }=\frac{4\varepsilon_{0}\varepsilon_{r}l_{\mathrm{in}}^{2}r_{\mathrm{in}}}{d_{0}^{2}}\sin(\frac{\varphi_{p}}{2})\theta =K_{c\theta }\theta$$
where $K_{cz},K_{cx},K_{c\theta }$ are the displacement-capacitance coefficients for every DOF, respectively. Formulas (4)–(6) demonstrate that when the proof mass position approaches the center of the electrode housing, the displacement detection circuit can convert the displacement into a linear change in differential capacitance.

## 4. Electrostatic Suspension Control Principle

The electrostatic suspension control model is used to characterize the effect of voltage on electrostatic force for each DOF near the equilibrium position. According to Table 2, the suspension voltage and electrostatic force are illustrated in Figure 7.

To minimize proof mass’surface charge coupled from surrounding electrodes, AC support schemes are employed here. Additionally, to further reduce the maximum suspension supplier, the preload voltage $U_{r}=V_{r}\sin\omega t$ and displacement measurement carrier $U_{c}=V_{m}\sin\omega_{m}t$ are applied to the proof mass via golden wire, while control voltages are applied to electrodes, as follows:(7)$$\begin{array}{l} U_{1}^{\prime }=-U_{1}=V_{bz}\sin\omega t\\ U_{\mathrm{out}4}=-U_{\mathrm{out}2}=V_{by}\sin\omega t\\ U_{\mathrm{in}4}=U_{\mathrm{in}4}^{\prime }=-U_{\mathrm{in}2}=-U_{\mathrm{in}2}^{\prime }=V_{b\theta }\sin\omega t \end{array}$$

When the proof mass generates displacement, neglecting second-order small quantities and high frequency terms, the electrostatic force can be obtained as follows:(8)$$\begin{array}{ll} F_{ez} & =F_{e1}-F_{e1}^{\prime }\\ & =\frac{\varepsilon_{0}\varepsilon_{r}A_{z}}{d_{0z}^{2}}V_{r}V_{bz}+\frac{\varepsilon_{0}\varepsilon_{r}(V_{r}^{2}+V_{bz}^{2}+V_{m}^{2})}{d_{0z}^{3}}z=H_{tz}V_{bz}+K_{ez}z \end{array}$$
(9)$$\begin{array}{ll} M_{\theta } & =2(F_{\mathrm{ein}2}+F_{\mathrm{ein}4}^{\prime }-F_{\mathrm{ein}4}+F_{\mathrm{ein}2}^{\prime })L_{M}\\ & =\varepsilon_{0}\varepsilon_{r}L_{M}[\frac{4\sin\frac{\varphi_{p}}{2}r_{\mathrm{in}}l_{\mathrm{in}}}{d_{0}^{2}}V_{r}V_{b\theta }+\frac{\left(\sin\varphi_{p}+\varphi_{p})r_{\mathrm{in}}l_{\mathrm{in}}^{2}(V_{r}^{2}+V_{b\theta }^{2}+V_{m}^{2}\right)}{2d_{0}^{3}}\theta ]=H_{M}V_{b\theta }+K_{M}\theta \end{array}$$
(10)$$\begin{array}{ll} F_{ey} & =F_{\mathrm{ein}4}+F_{\mathrm{ein}4}^{\prime }+F_{\mathrm{eout}2}-F_{\mathrm{ein}2}-F_{\mathrm{ein}2}^{\prime }-F_{\mathrm{eout}4}\\ & =\frac{2\varepsilon_{0}\varepsilon_{r}r_{\mathrm{out}}l_{\mathrm{out}}\sin\frac{\varphi_{p}}{2}}{d_{0}^{2}}V_{r}V_{by}+\frac{\varepsilon_{0}\varepsilon_{r}(\varphi_{p}+\sin\varphi_{p})[(V_{r}^{2}+V_{by}^{2}+V_{m}^{2})r_{\mathrm{out}}l_{\mathrm{out}}+2(V_{r}^{2}+V_{b\theta }^{2}+V_{m}^{2})r_{\mathrm{in}}l_{\mathrm{in}}]}{2d_{0}^{3}}y=H_{ty}V_{by}+K_{ey}y \end{array}$$
where $H_{t}$ is the electrostatic force coefficient and $K_{e}$ is the negative stiffness coefficient. The Formulas (8)–(10) demonstrate that the electrostatic force generated by the suspension voltage can be accurately described by a linear model near the equilibrium position.

## 5. Controller Design

The initial levitation of the proof mass and suspension in the nominal gap will be specified, respectively, in this section.

### 5.1. Control Method of Initial Levitation Process

During the initial levitation process, single-plate suspension is most efficient to lift up the proof mass from the stop pin. The levitation process exclusively operates in the translational degree of freedom (DOF) to prevent disturbances caused by severe cross-talk during initial levitation. The amplitude of the suspension voltage applied to the electrode plate is constant as $V_{r}$ during whole initial levitation process. This maximizes the electrostatic force generated by the upper electrode plate, placing the control system in a one-plate support mode. As the nonlinearity becomes less pronounced, the single-plate suspension transitions to a double-plate suspension.

### 5.2. Control Method of Suspension Stage

In the suspension stage, double-plate suspension is employed to maintain the system’s robustness. The overall system diagram of the single DOF electrostatic suspension accelerometer is shown in Figure 8 where $G_{s}$, $G_{b}$ and H represent gain or sensitivity for the displacement detection circuit, high-voltage amplification circuit, and electrostatic actuator, respectively. The design steps for the Z DOF controller are introduced in the following sentences, and the controllers for the other DOFs can be designed using a similar approach. Additionally, to ensure the robustness of the controller against cross-coupling and large inputs, we opted to select the maximum value of $K_{e}$ within the operating range when designing the controllers for each DOF.

The electrostatic suspension open-loop transfer function is:(11)$$G_{\mathrm{open}}(s)=\frac{G_{s}(s)G_{\mathrm{ctrl}}(s)G_{b}(s)H}{ms^{2}-K_{e}}$$

Obviously, the negative stiffness makes it an unstable system [22], and after introducing $G_{\mathrm{ctrl}}(s)$, the closed-loop transfer function is:(12)$$G_{\mathrm{close}}(s)=\frac{G_{s}(s)G_{\mathrm{ctrl}}(s)G_{b}(s)H}{ms^{2}-K_{e}+G_{s}(s)G_{\mathrm{ctrl}}(s)G_{b}(s)H}$$

The support stiffness is important for the reliability of the ESA. The support stiffness of the electrostatic suspension system is the inverse of the transfer function from the external perturbation force to the displacement response, which is expressed as:(13)$$K_{x-f}(s)\overset{def}{=}\left|\frac{ma}{x(s)}\right|=\left|ms^{2}-K_{e}+G_{s}(s)G_{\mathrm{ctrl}}(s)G_{b}(s)H\right|$$

The stiffness characteristics of the system curve form a basin shape with a low center and high sides, and the mid-frequency stiffness mainly depends on the gain of the system controller. Static position error, stability, and robustness are the main concerns for controller design. To ensure the dynamic stiffness and stability margin of the system, a corner frequency $\omega_{c}=2\pi \times 420\;\mathrm{rad}/s$ is chosen, along with a phase margin of $\phi_{M}=47^{\circ }$.

As Formula (14) shows, a proportional-integral-phase-lead (PIDPL) network is employed as a position loop controller, effectively attenuating high-frequency noise stemming from the pure differentiation of PID.
(14)$$G_{\mathrm{ctrl}}(s)=K_{c}(1+\frac{1}{T_{i}s})\frac{(T_{d}+\tau_{d})s+1}{T_{d}s+1}$$

Denoting $\gamma =(T_{d}+\tau_{d})/T_{d}$, the maximum phase lead $\phi_{\max}$ and corresponding angular frequency $\omega_{\max}$ in a phase-lead network can be expressed as:(15)$$\begin{array}{l} \phi_{\max}=\arctan\frac{\sqrt{\gamma }-1/\sqrt{\gamma }}{2}\\ \omega_{\max}=\frac{1}{\sqrt{T_{d}(T_{d}+\tau_{d})}} \end{array}$$

Given $\omega_{\max}=\omega_{c}$ and γ=12, solving the Formula (15), the parameters $T_{d}$ and $\tau_{d}$ of the phase lead network can be determined. The value of γ is recommended to be within the range of 6 to 12, and the specific values can be further determined through experimental observation.

At corner frequency $\omega_{c}$, the magnitude and phase angle of the frequency characteristics of the open-loop system are expressed as:(16)$$\begin{array}{l} \left|K_{c}(1+\frac{1}{j\omega_{c}T_{i}})\right|\cdot \left|\frac{G_{s}(s)G_{b}(s)H(s)}{-m\omega_{c}^{2}-K_{e}}\frac{(T_{d}+\tau_{d})j\omega_{c}+1}{T_{d}j\omega_{c}+1}\right|=1\\ \arg[K_{c}(1+\frac{1}{j\omega_{c}T_{i}})]+\arg[\frac{G_{s}(s)G_{b}(s)H(s)}{-m\omega_{c}^{2}-K_{e}}\frac{(T_{d}+\tau_{d})j\omega_{c}+1}{T_{d}j\omega_{c}+1}]=-180^{\circ }+\phi_{M} \end{array}$$

Based on the design values of the ESA and the parameters of the phase lead network, combined with Formula (16), the values of $K_{c}$ and $T_{i}$ can be calculated. Finally, the designed controller is:(17)$$G_{\mathrm{ctrl}}(s)=54,144(1+\frac{1}{0.0034s})\frac{(0.000109+0.0012)s+1}{0.000109s+1}$$

The Bode diagram of the open-loop system after adding the controller is shown in Figure 9.

As shown in Figure 9, the corner frequency of the system is approximately 420 Hz, and the phase margin is about 47°, which is satisfied for the ESA control.

What should be noted is that the purely integral link in the controller often introduces an unbearable error when saturation occurs. This necessitates a prolonged reversal of the control error for the system to resume normal operation. Consequently, the system experiences a notable transient process, resulting in large and unstable low-frequency oscillations, commonly known as “winding”. The winding phenomenon also occurs during the transition from the initial levitation process to the suspension stage of the ESA.

To mitigate integrator winding during the transition, a backward computation and tracking branch are incorporated alongside the PIDPL controller. This backward branch dynamically resets the integrator with a time constant. In cases of output saturation, it recalculates the new output to align with the saturation limit. This approach helps withstand significant disturbances and facilitates a smooth transition from the initial levitation process to the suspension stage. Figure 10 depicts a block diagram of a PIDPL controller with inverse “winding”.

The additional feedback branch consists of two parts: the feedback error signal $e_{s}$ and the negative feedback gain $1/T_{t}$. The feedback error signal represents the disparity between the input and output of the actuator. When the system operates within the linear region, the feedback error signal is zero, ensuring the system operates as intended. When the actuator is saturated, the feedback branch around the integrator works. At steady state, the output of the integrator is adjusted to render its input null, as follows:(18)$$\frac{K_{c}}{T_{i}}e(t)-\frac{1}{T_{t}}e_{s}(t)=0$$
where, e(t) is the output of the phase-lead network. Hence, it is obtained that:(19)$$u(t)=v(t)-\frac{T_{t}K_{C}}{T_{i}}e(t)$$

It shows that the additional feedback branch consistently attenuates the control variable, guiding it back to the linear region, and thereby preventing “winding” of the integrator.

The controller output reset rate is governed by the feedback gain $1/T_{t}$. The tracking time constant $T_{t}$ should be kept relatively small to facilitate a faster integrator reset. However, combined with differential correction, selecting a value that is too small may introduce spurious signals, leading to output saturation and incorrect integrator settings. As a general rule, it should be greater than the differential time constant $T_{d}+\tau_{D}$ yet remain less than the integral time constant $T_{i}$. An intermediate value is generally suggested as follows:(20)$$T_{t}=\sqrt{T_{i}(T_{d}+\tau_{D})}=0.00211$$

### 5.3. Results of Simulation

The dynamic characteristics of the system are further verified by simulation. A MATLAB simulation model based on Simulink is given in Figure 11.

The Simulink model comprises five main components: the mass module, displacement detection module, controller module, voltage amplifier module, and electrostatic actuator module. In the mass module, electrostatic force and disturbances act on the system, resulting in displacement. This displacement is detected by the displacement detection module, which generates voltage signals proportional to it. These voltage signals are then fed into the controller module for signal processing and the generation of control signals. Subsequently, the voltage amplifier module comes into play, amplifying and converting the control signals into support voltages that are applied to the electrode plates. Finally, the electrostatic actuator module utilizes these support voltages and displacements of proof mass to calculate the magnitude of the electrostatic force.

To compare the effect of inverse “winding”, the displacement changes during the initial levitation process are simulated, respectively, with PIDPL alone and PIDPL with inverse winding as shown in Figure 12.

As shown in Figure 12a, the initial levitation process has undergone a vigorous oscillation adjustment process at Z direction with the PIDPL controller, even failing to stabilize at X and Y directions. Nevertheless, with the incorporation of the inverse “winding” feedback branch, we successfully completed the initial levitation process and ensured a stable transition from the initial levitation to the suspension stage. By reducing the integral term during the initial support process, the inverse “winding” feedback branch facilitates a shorter and more stable transient process for the control system.

At the suspension stage, a step disturbance acceleration of 1 g magnitude in Z and X & Y DOF is applied, as is a step angular acceleration with a magnitude of $10{\;\mathrm{rad}/s}^{2}$ in θ & ϕ DOF. It is worth noting that Figure 13 presents the results obtained when external disturbances are applied near the equilibrium position, assuming that the proof mass has completed the levitation process with its initial position at the zero point. Figure 12, on the other hand, shows the results obtained when the initial position is far from the equilibrium position, overcoming gravity during the levitation process. According to the simulation results presented in Figure 13, the displacement peak value is less than 4.5 μm under ±1 g disturbance, and the static position error is less than 10 nm in translational DOF. The rotation angle peak value is less than $4\times 10^{-6\;}\mathrm{rad}$ under $\pm 10{\;\mathrm{rad}/s}^{2}$ disturbance, and the static position error is less than $1\times 10^{-8}\;\mathrm{rad}$ in rotational DOF.

Additionally, we meticulously conducted a simulation experiment to test the robustness and boundaries of reliable control. We examined the experimental situation when the parameters nominal gap, voltage amplification gain, and displacement detection circuit gain each had errors of ±5% and ±10%. These three variables are challenging to determine accurately in actual experiments and significantly impact the specific values of electrostatic negative stiffness, actuator system gain, and detection system gain. The step response and initial levitation process are, respectively, iteratively simulated for all 125 total situations, as shown in Figure 14 and Figure 15.

According to the results, when the primary error source has an error range of less than 10%, the controller can reliably execute the support and levitation functions.

## 6. Conclusions

In this paper, we present a design for a ground-based ESA. The ground-used scenery put forward high requirements for measuring range (>1g) and wider dynamic range (>10 Hz). To compromise high suspension voltage and high resolution, a proof-mass hollow cylinder with a thin flange fixed in the center is put forward due to its large surface area-to-mass ratio. The corresponding 5-DOF electrode distribution and control schemes are discussed. The electrodes are strategically distributed on the inner and outer surfaces of the proof mass cylinder structure to maximize surface area utilization and reduce the voltage amplitude required for support.

The displacement detection model and electrostatic force model are detailed and explained in the present operation system of the ground-based ESA. We employ a single plate support scheme in the initial levitation process to simplify the control scheme and eliminate cross-coupling. In the suspension stage, in order to resist large disturbances and ensure a stable transition from the initial levitation process to the suspension stage, we employ a PIDPL network controller with an inverse “winding” feedback branch. Simulation results show that a swift and stable initial levitation process with no obvious overshooting is achieved. Moreover, in the suspension stage, the simulation results show that the peak value is less than 4.5 μm under ±1g disturbance, and the static position error is less than 10 nm. The rotation angle peak value is less than $4\times 10^{-6}\;\mathrm{rad}$ under ${\pm 10\;\mathrm{rad}/s}^{2}$ disturbance, and the static position error is less than $1\times 10^{-8}\;\mathrm{rad}$ in rotational DOF.

## Figures and Tables

**Figure 1 sensors-24-04029-f001:**
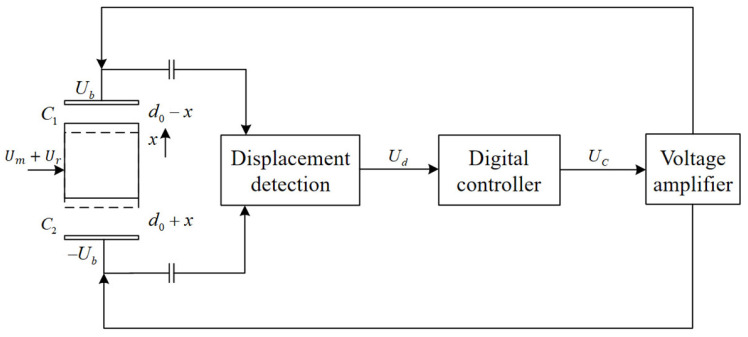
Schematic sketch of electrostatic suspension system overview and operation principle in one DOF.

**Figure 2 sensors-24-04029-f002:**
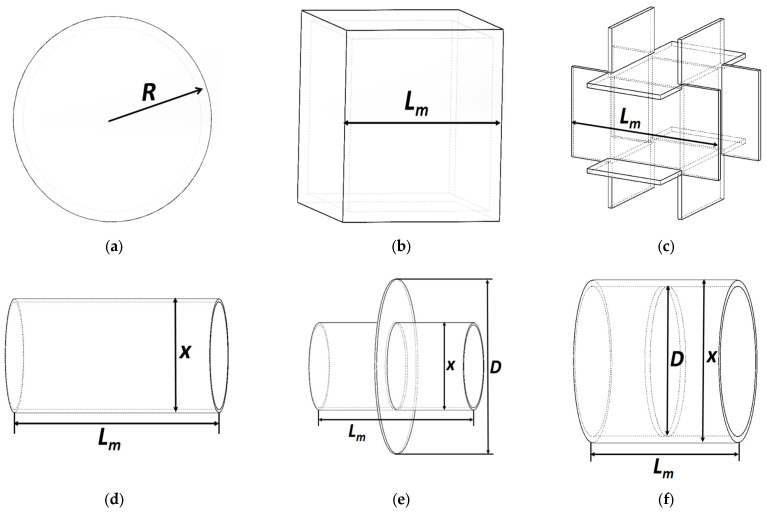
Typical proof mass structural diagrams: (**a**) hollow sphere; (**b**) hollow hexahedron; (**c**) six thin hollow plates; (**d**) hollow cylinder; (**e**) hollow cylinder with outer flange; (**f**) hollow cylinder with inner flange.

**Figure 3 sensors-24-04029-f003:**
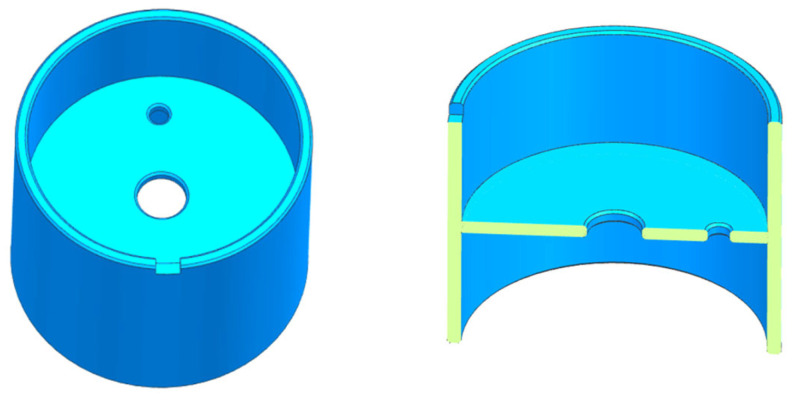
Schematic Diagram of the Proof Mass Structure.

**Figure 4 sensors-24-04029-f004:**
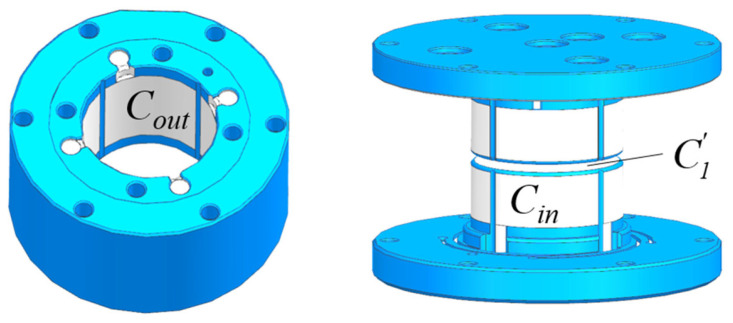
Schematic Diagram of Electrode Structure.

**Figure 5 sensors-24-04029-f005:**
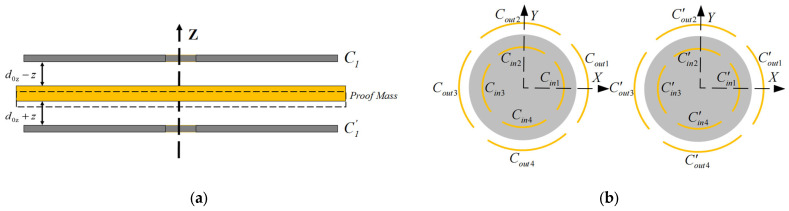
The numbering definition pf electrodes: (**a**) Numbering definition of planar electrodes in disk direction; (**b**). The numbering definition of cylindrical electrodes in the cylinder direction.

**Figure 6 sensors-24-04029-f006:**
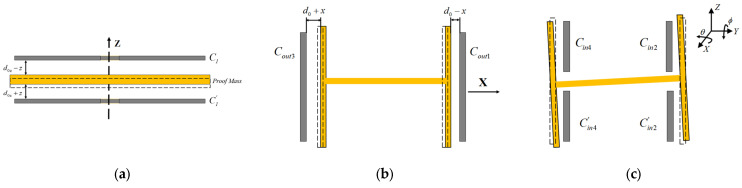
Schematic diagram of displacement detection: (**a**) displacement detection in Z DOF; (**b**) displacement detection in X & Y DOF. (**c**) displacement detection in θ & ϕ DOF.

**Figure 7 sensors-24-04029-f007:**
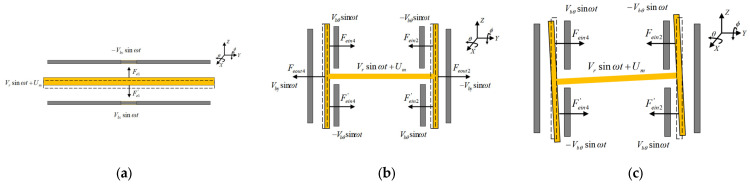
Schematic diagram suspension control principle: (**a**) voltage load and electrostatic force suspension scheme in Z DOF; (**b**) voltage load and electrostatic force suspension scheme in X & Y DOF. (**c**) voltage load and electrostatic force suspension scheme in θ & ϕ DOF.

**Figure 8 sensors-24-04029-f008:**
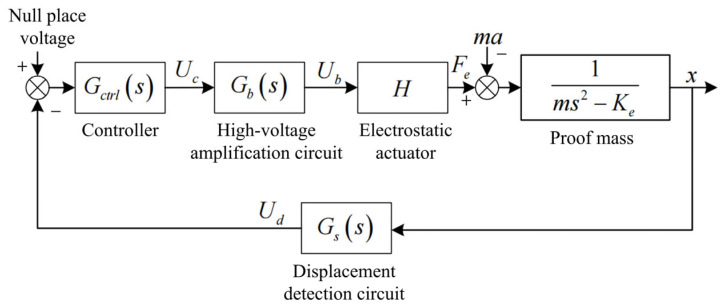
Schematic diagram of a single-degree-of-freedom electrostatic suspension.

**Figure 9 sensors-24-04029-f009:**
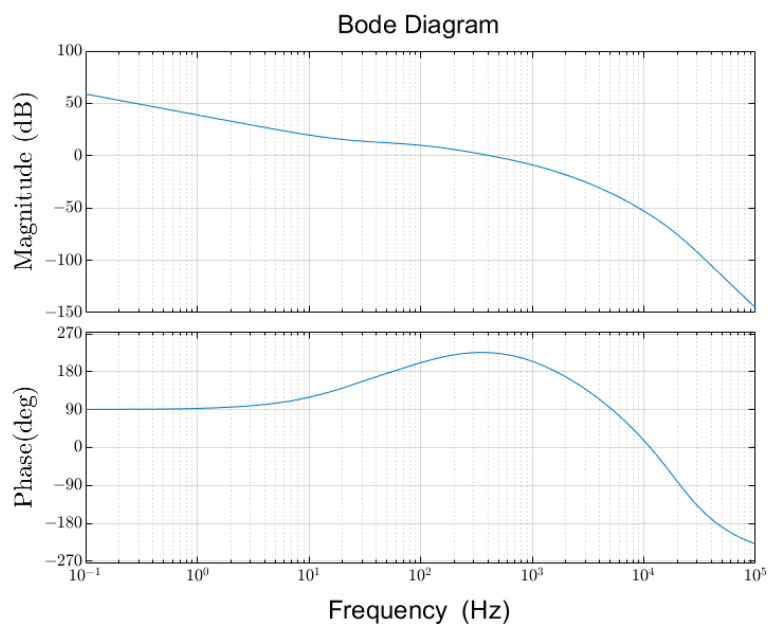
Bode diagram of the open-loop system.

**Figure 10 sensors-24-04029-f010:**
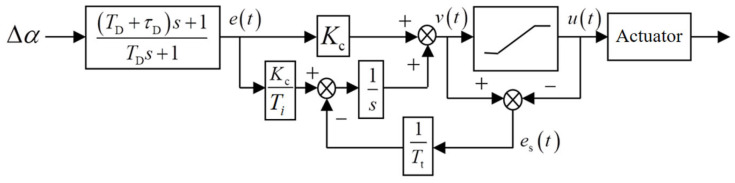
Controller with inverse “winding”.

**Figure 11 sensors-24-04029-f011:**
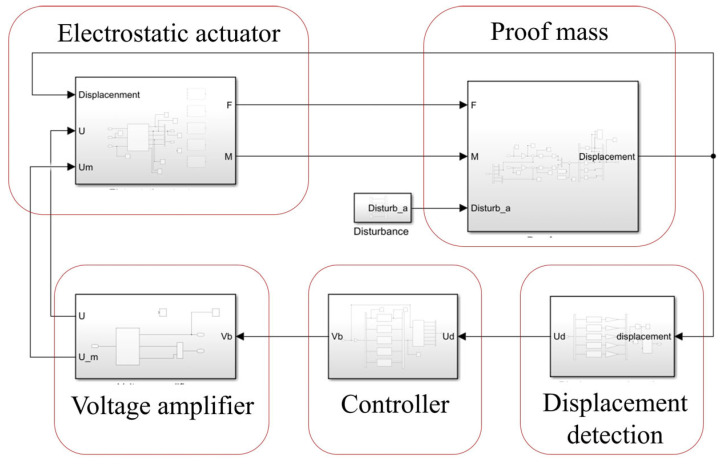
Simulink model of system structure.

**Figure 12 sensors-24-04029-f012:**
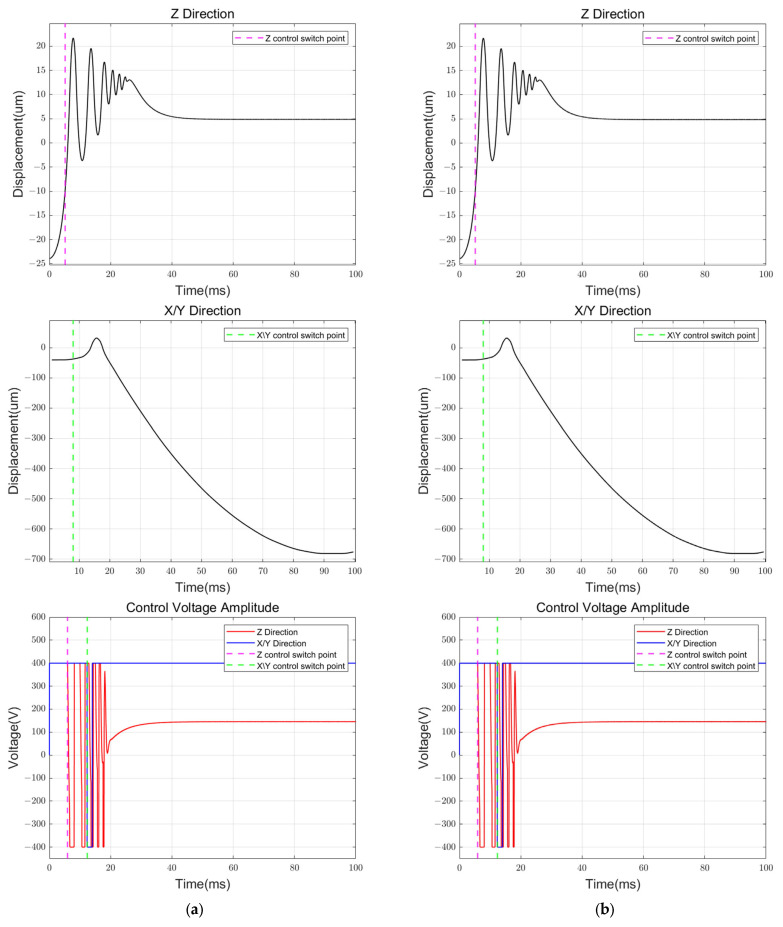
Simulink simulation results of the initial levitation process: (**a**) PIDPL alone; (**b**) PIDPL with inverse “winding”.

**Figure 13 sensors-24-04029-f013:**
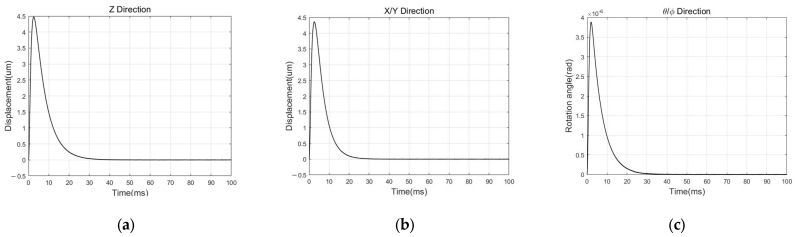
Simulink simulation results of the suspension stage: (**a**) result in Z DOF; (**b**) result in X & Y DOF. (**c**) result in θ & ϕ DOF.

**Figure 14 sensors-24-04029-f014:**
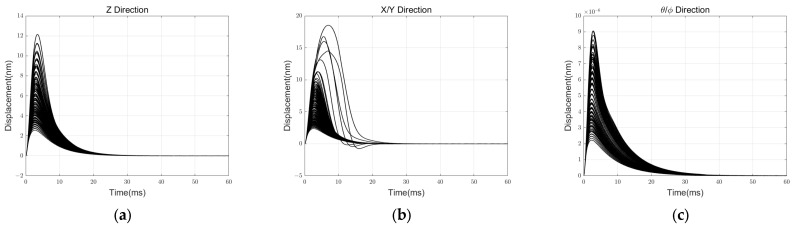
Step response simulation results in a robust analysis: (**a**) result in Z DOF; (**b**) result in X & Y DOF. (**c**) result in θ & ϕ DOF.

**Figure 15 sensors-24-04029-f015:**
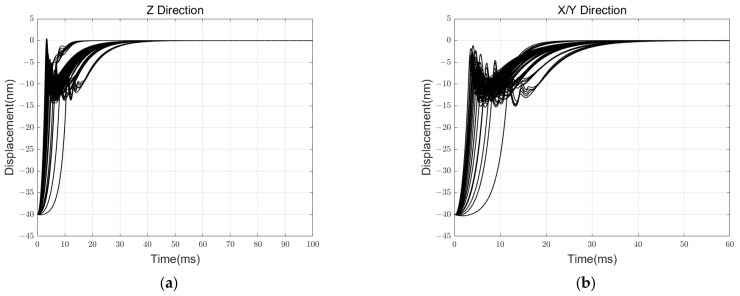
Initial levitation simulation results in a robust analysis: (**a**) result in Z DOF; (**b**) result in X & Y DOF.

**Table 1 sensors-24-04029-t001:** Comparison of surface area to mass ratio of typical proof mass structures *^1^.

Poof Mass Structure	Surface Area Formula	Surface Area Value $\left(\times R^{2}\right)$	Mass Formula	$\mathrm{Quality}\;\mathrm{Value}\;(\times R^{3})$	$\mathrm{Surface}\;\mathrm{Area}\;\mathrm{to}\;\mathrm{Mass}\;\mathrm{Ratio}\;(\times R^{-1})$
Hollow sphere (Radius R)	$4\pi R^{2}$	12.57	$\frac{4\pi }{3}R^{3}-\frac{4\pi }{3}{\left(R-b\right)}^{3}$	0.60	21.0 *^2^
Hollow hexahedron (side length $L_{m}$)	$6L_{m}^{2}$	15.59	$L_{m}^{3}-{\left(L_{m}-2b\right)}^{3}$	0.73	21.3 *^2^
Six thin, hollow plates $\left(\begin{array}{l} L_{m}\times \frac{L_{m}}{2}\\ L_{m}=1.612R \end{array}\right)$ (side length $L_{m}$)	$3L_{m}^{2}$	7.80	$3L_{m}^{2}b$	0.39	20.0 *^2^
Hollow cylinder $\left(\begin{array}{l} x=\frac{L_{m}}{2}\\ L_{m}=2.77R \end{array}\right)$	$2\pi (x-b)L_{m}$	23.23	$\frac{[x^{2}-{\left(x-2b\right)}^{2}]\pi L_{m}}{4}$	0.58	40.1
Hollow cylinder with an outer flange $\left(\begin{array}{l} x=\frac{L_{m}}{2}\\ D=L_{m}\\ L_{m}=2.72R \end{array}\right)$	$\begin{array}{l} 2\pi (x-b)L_{m}\\ +\frac{\pi (D^{2}-x^{2})}{2} \end{array}$	31.10	$\frac{[x^{2}-{\left(x-2b\right)}^{2}]\pi L_{m}}{4}+\frac{(D^{2}-x^{2})\pi b}{4}$	0.78	39.9
Hollow cylinder with an inner flange $\left(\begin{array}{l} x=L_{m}\\ D=x-2b\\ L_{m}=1.75R \end{array}\right)$	$\begin{array}{l} 2\pi (x-b)L_{m}\\ +\frac{\pi D^{2}}{2} \end{array}$	22.97	$\frac{[x^{2}-{\left(x-2b\right)}^{2}]\pi L_{m}}{4}+\frac{\pi bD^{2}}{4}$	0.57	40.3

*^1^ All structures have equal volumes. When calculating mass and surface area, a density value of 1 and a thin wall thickness of 0.05R are used. *^2^ Because the internal space is isolated from the external space, it is not possible to arrange fixed electrode plates within the internal space, and only its outer surface area is considered when calculating the surface area-to-mass ratio.

**Table 2 sensors-24-04029-t002:** The symbols for the design parameters.

Design Parameters	Symbols
The inner diameter of the outer cylindrical electrode	$r_{\mathrm{out}}$
The length of the outer cylindrical electrode	$l_{\mathrm{out}}$
The included angle of the inner and outer cylindrical electrodes	$\varphi_{p}$
The diameter of the inner cylindrical electrode	$r_{\mathrm{in}}$
The length of the inner cylindrical electrode	$l_{\mathrm{in}}$
The surface area of the disk electrode in the Z-direction	$A_{z}$
The gap between the electrode and the proof mass	$d_{0}$
The electrostatic force arm length is for rotational freedom	$L_{m}$
The capacitance of the outer cylindrical electrode	$C_{\mathrm{out}}$
The capacitance of the inner cylindrical electrode	$C_{\mathrm{in}}$
The capacitance of the disk electrode in the Z-direction	$C_{z}$
The mass of the proof mass	m

**Table 3 sensors-24-04029-t003:** Electrodes detection and force application allocation scheme.

Degree of Freedom	Electrodes for Displacement Detection and Suspension Voltage Application
X	$C_{\mathrm{out}1},C_{\mathrm{out}3}$
Y	$C_{\mathrm{out}2},C_{\mathrm{out}4}$
Z	$C_{1},C_{1}^{\prime }$
ϕ	$C_{\mathrm{in}1}//C_{\mathrm{in}3}\prime ,C_{\mathrm{in}3}//C_{\mathrm{in}1}\prime$
θ	$C_{\mathrm{in}2}//C_{\mathrm{in}4}\prime ,C_{\mathrm{in}4}//C_{\mathrm{in}2}\prime$

## Data Availability

The original contributions presented in the study are included in the article, further inquiries can be directed to the corresponding author.

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
