# Peer review of "A Ground-Based Electrostatically Suspended Accelerometer"

_sensors, 2024, doi:10.3390/s24124029_

Round 1

Reviewer 1 Report

Comments and Suggestions for Authors

In this paper, the authors have designed an electrostatically suspended accelerometer for applications in marine and airborne gravity measurement and gravity-assisted navigation. It presents a proof mass structure with a significant area-to-mass ratio,to reduce the voltage for suspension and employs a "PIDPL" controller to suppress integrator overshoot.  

There are some specific questions as follows.

i. The authors in Section 2.2 analyzed the face-to-mass ratio of six types of proof mass to obtain the minimum equilibrium voltage. However, Six thin hollow plates only calculate the unilateral area, whereas Hollow cylinders calculate both the inner and outer cylindrical surfaces, and what is the difference in the evaluation criteria. the face-to-mass ratio for three orthogonal thin hollow plates structures are suggested to be given.

ii. The authors use a pin-hole to constrain rotational DOF, according to the design, when the proof mass is suspended, the rotational DOF alone the axis-Z is not controlled. Then, will the pin collide with the hole to affect the control effect of the proof mass.

iii. In order to simplify the control strategy and reduce cross-talk, the detection and force application electrodes for each DOF are independent of each other. This scheme will reduce the utilization of electrode area and increase the voltage. Since AC support schemes are employed, the multiple orthogonal frequency AC control used in space gravitational wave detection may be a better choice.

iv. The authors simulate the control parameters of the proof mass in Section 5, but the initial levitation process simulation in Figure 12 does not reflect the process from a one-plate support to double-plate suspension. It is suggested to consider the phenomenon of sudden changes in the pole-plate voltage introduced due to state transitions.

v. It is suggested that the paper add a table to summarize the parameters and indicators.

Reviewer 2 Report

Comments and Suggestions for Authors

Major comments

1- The control design method looks quite classical, but seems to work at least in the nominal setup. In Figure 8, you show a DOF design interconnection. Does it mean you consider all DOFs decoupled for control design? And if so, could you please leave that clearer in the text? For instance, specify how many linear PIDPL controllers you obtained in this process.

2- Regarding Section 5.3, it is not clear whether there is one single simulator for the system, or you have one simulator per DOF as indicated in Figure 11 (z axis). This must be clarified. Also in the case there is one single simulator, you should clarify the implementation/modelling of cross-couplings. A disturbance step in one coordinate, should have an effect on the others due to the actuation and sensing, which are generally coupled for this kind of systems.

3- Also related to Section 5.3, the results show overall good performance in a nominal setup. But what happens when there are deviations on the system or in the actuation system? For instance, the stiffness coefficient is generally difficult to compute/estimate. I strongly suggest showing how the system performs when there are parametric uncertainties in a Monte Carlo analysis. This could reveal the robustness of the control solution and how the performance degrades due to unknowns in the modelling.

Minor comments

* There are elements in some equations that are not properly presented. For instance, Eq(1) epsilon_0, epsilon_r, Az. Please check this throughout the text.

* In page 5, the authors present a novel proff mass structure. They state that they have a high surface area to mass ratio. Which is this number and how does it compare with the values shown in Table 1?

* In page 6 (line 150), the caption shows that figures 6b and 6c are for two DOF, but the figures represent only X and theta, respectively. Maybe the authors should clarify that the scheme for Y is the same as in Figure 6b, and similarly with phi as in Figure 6c.

* In page 9 (line 218), the authors suggest using gamma = 12. Why is that? Is this a design choice? If so, please elaborate.

* In page 9 (Eq. 16), there are missing elements in the lower part of Eq 16. I guess a = arg and -1 = -180 deg

* In page 11 (Figure 11). Could the authors justify why the mass block output is feedback to the electrostatic actuator? This should be clarified explicitly in the text.

Please check the following cross-references:

* Page 8 (line 197): [...] is shown in Figure '8' where [...]

* Page 12 (line 285): As shown in Figure '12'.a, 't'he initial levitation [...]

Comments on the Quality of English Language

There some typos along the text that must be corrected:

* Page 1 (line 37):  [...] waves detection [3], 'and' the test of geodetic [...]

* Page 2 (line 87): [...] operation principle in '1' DOF

* Page 10 (line 252): When the actuator is saturate'd' [...]

* Page 12 (line 283): (a) PIDPL alon'e'

* Page 12 (line 291): [...] facilitates a shorter and more 's'table transient [...]

Round 2

Reviewer 2 Report

Comments and Suggestions for Authors

The authors applied all the changes required by this reviewer.